# *Bacillus megaterium*-Mediated Synthesis of Selenium Nanoparticles and Their Antifungal Activity against *Rhizoctonia solani* in Faba Bean Plants

**DOI:** 10.3390/jof7030195

**Published:** 2021-03-09

**Authors:** Amr H. Hashem, Amer M. Abdelaziz, Ahmed A. Askar, Hossam M. Fouda, Ahmed M. A. Khalil, Kamel A. Abd-Elsalam, Mona M. Khaleil

**Affiliations:** 1Botany and Microbiology Department, Faculty of Science, Al-Azhar University, Cairo 13759, Egypt; amr.hosny86@azhar.edu.eg (A.H.H.); drahmed_askar@azhar.edu.eg (A.A.A.); hossamfouda2016@azhar.edu.eg (H.M.F.); ahmed_khalil@azhar.edu.eg (A.M.A.K.); 2Biology Department, College of Science, Taibah University, Yanbu 41911, Saudi Arabia; mkhaleil@taibahu.edu.sa; 3Plant Pathology Research Institute, Agricultural Research Center (ARC), Giza 12619, Egypt; 4Botany and Microbiology Department, Faculty of Science, Zagazig University, Zagazig 44519, Egypt

**Keywords:** *Vicia faba*, plant disease, root rot, *R. solani*, Se-NPs, nano-biosynthesis, plant promotion

## Abstract

Rhizoctonia root-rot disease causes severe economic losses in a wide range of crops, including *Vicia faba* worldwide. Currently, biosynthesized nanoparticles have become super-growth promoters as well as antifungal agents. In this study, biosynthesized selenium nanoparticles (Se-NPs) have been examined as growth promoters as well as antifungal agents against *Rhizoctonia solani* RCMB 031001 in vitro and in vivo. Se-NPs were synthesized biologically by *Bacillus megaterium* ATCC 55000 and characterized by using UV-Vis spectroscopy, XRD, dynamic light scattering (DLS), and transmission electron microscopy (TEM) imaging. TEM and DLS images showed that Se-NPs are mono-dispersed spheres with a mean diameter of 41.2 nm. Se-NPs improved healthy *Vicia faba* cv. Giza 716 seed germination, morphological, metabolic indicators, and yield. Furthermore, Se-NPs exhibited influential antifungal activity against *R. solani* in vitro as well as in vivo. Results revealed that minimum inhibition and minimum fungicidal concentrations of Se-NPs were 0.0625 and 1 mM, respectively. Moreover, Se-NPs were able to decrease the pre-and post-emergence of *R. solani* damping-off and minimize the severity of root rot disease. The most effective treatment method is found when soaking and spraying were used with each other followed by spraying and then soaking individually. Likewise, Se-NPs improve morphological and metabolic indicators and yield significantly compared with infected control. In conclusion, biosynthesized Se-NPs by *B. megaterium* ATCC 55000 are a promising and effective agent against *R. solani* damping-off and root rot diseases in *Vicia faba* as well as plant growth inducer.

## 1. Introduction

The global population will increase to about eight billion people in 2025 and nine billion people in 2050, which requires an increase in agricultural production to feed a rapidly expanding world population [1]. Unfortunately, food security is threatened by crop losses due to attacks of pathogens, including fungi [2,3], and it is estimated that around one-third of the global crop is lost each year due to plant diseases [4]. Phytopathogenic fungi cause losing crop-yield (20–40%) annually worldwide [5]. *Vicia faba* is the main important economic legume over the world that is used as human food, livestock fodder, and silage production [6]. In Egypt, *Vicia faba* (Faba Bean) is one of the most important economic legume crops as a source of protein (18–32%), carbohydrates (55–63%), minerals (2–3.5%), fat (0.5–5.6%), phosphorus, iron, calcium, and vitamins in food [7]; also, it has an ecological role in improving soil quality by the nitrogen fixation and enhances N and P nutrition of cereals [8]. Generally, *Vicia faba* plays a vital role in crop rotation and limiting the disease cycles of various plant pathogens. Unfortunately, *Vicia faba* suffers from many abiotic and biotic stresses that have reduced crop production and led to a decrease in the cultivated area of bean plants around the world from 5 million in 1965 to 2.4 million hectares in 2016. It is susceptible to soilborne fungal pathogens, including *Rhizoctonia solani*, which causes serious root rot disease that harms the quality and quantity of crop yield [9,10,11,12,13], causing a significant gap between production and consumption of *Vicia faba* in Egypt [6]. Moreover, *R. solani* has a broad host range including solanaceous crops, cereals, fruits and vegetables such as potatoes, cucumbers, eggplant, peppers, sugar beet, lettuce, tomatoes, and melon, cotton, and forest trees for a long time [14,15]. *R. solani* is an aggressive fungal plant pathogen with a highly resistant structure called sclerotia, which allows the fungus to survive under environmental conditions [15]. Although fungicides are effective for controlling *R. solani*, they pollute the environment, have a high cost, and also affect other beneficial organisms in the soil [16]. Fungi are the largest group among microbes, where are used in different applications as nanotechnology, bioremediation, bio-deinking, food products, enzyme production, organic acids, and biofuels [17,18,19,20,21,22,23]. Dong et al. [24] reported that the management of plant diseases can be achieved by Gly-Cu(OH)_2_ NPs by reducing the phytotoxicity to plants and improving the utilization of copper-based bactericides. Krutyakov et al. [25] proved that silver nanoparticles are an effective agent for increasing yields as well as decreasing plant diseases besides having a low harmful effect on humans and animals. The application of nanoparticles in agriculture is beneficial for improving the growth and yield of crops as well as inhibiting plant pathogens [26] by facilitating the uptake of macromolecules needed to increase resistance to plant diseases and promote growth [27]. The biological synthesis of metal nanoparticles provides an eco-friendly and cost-effective method. An alternative approach to the synthesis of metal nanoparticles is to apply biomaterials such as plants, microorganisms encompassing groups such as bacteria, yeasts, fungi, and actinomycetes as manufactories [28]. Ag-NPs can be utilized as a management and control agent against various fungal diseases of plants especially *Rhizoctonia solani* and have antifungal activity against mycelium as well as sclerotia [29]. Selenium nanoparticles (Se-NPs) synthesized from a biological source has been shown to have antimicrobial activity against pathogenic microorganisms including fungi [30]. Se-NPs is suggested to be used as a fungicide in agriculture because it has the advantage of being less toxic to humans and animals than synthetic fungicides [31]. In the same context, selenium is an essential trace element for plants growth. It is usually involved in coenzyme activation and physiological facilitation in crop plants, which contributes to food production and quality [32]. In our understanding, bacteriogenic Se-NPs antifungal action against Rhizoctonia diseases of faba bean plants is not thoroughly studied. The main aim of the current research is to (1) biosynthesize Se-NPs by *B. megaterium,* (2) characterize the physicochemical properties of the produced nanoparticles by UV-Vis spectroscopy, XRD, dynamic light scattering (DLS), and transmission electron microscopy (TEM) imaging, (3) assess and evaluate the antifungal activity of Se-NPs against *Rhizoctonia* RCMB 031001 root rot of *Vicia faba* in vitro and in vivo, (4) analyze photosynthetic pigments, metabolic indicators, protein, and phenolics compounds of *Vicia faba*, and (5) understand the antifungal mechanisms and the effects of Se-NPs on oxidative enzymes such as polyphenol oxidase (PPO) and peroxidase (POX) in *Vicia faba* under pot conditions using assays.

## 2. Materials and Methods

### 2.1. Biosynthesis of Se-NPs

Se-NPs were produced using *Bacillus megaterium* culture supernatant (as reducing and stabilizing agents). Bacteria were subcultured on nutrient broth media in conical flasks and incubated with shaking aerobically at 37 °C for 48 h. After an incubation period, the bacterial cells were removed from the suspension by filtration through a 0.44 µm PVDF filter; then, they were centrifuged at 10,000 rpm to remove occasional bacterial cells and macromolecules [33]. The next step was mixing cell-free supernatant with the selenious acid suspension (1 mM) by quotient (1:1) *v*/*v*. The mixtures were stirred at a controlled room temperature of about 25 °C. The process of selenious acid reduction was monitored by color change of the cell-free supernatant from colorless to reddish color [22,23,24,25]. The suspension of SeNPs was further centrifuged at 12,000 rpm for 30 min, and the collected precipitate pellet was dried and weighed. The concentration was calculated as follows: 1 mg of SeNPs was dissolved in 1 mL of DMSO, where the final concentration was 1000 µg/mL.

### 2.2. Characterization of Se-NPs

The characterization of Se-NPs was performed by using JASCO V-560, UV-Vis. spectrophotometer, Tokyo, Japan, at the wavelength range from 200–900 nm and at a resolution of 1 nm. Cell-free supernatant without SeO_2_ was used as blank to adjust the baseline. Toward particle size investigation, the specimens were diluted ten times by deionized water before being estimated. To determine the morphology and size of the manufactured Se-NPs, TEM microscopy, model JEOL JEM-100 CX (Peabody, MA, USA) was used. TEM imaging was carried out by drop covering the Se-NPs upon carbon-coated TEM layers. Dynamic light scattering (DLS) was used to determine the size distribution, while the average particle size was determined by PSSNICOMP 380-ZLS particle sizing system (St. Barbara, CA, USA). For XRD analysis, the adjusted sample was centrifuged, and the precipitate was dried under vacuum and taken for XRD analysis. X-ray diffraction patterns were obtained with XRD- 6000 series, including stress analysis, residual austenite quantitation, crystallite size/lattice strain, crystallinity calculation, and materials analysis via overlaid X-ray diffraction patterns Shimadzu apparatus using nickel-filter and Cu-Ka target, Shimadzu Scientific Instruments (SSI), (Kyoto, Japan). The average crystalline size of the Se-NPs was also determined by using Debye–Scherrer equation: D = kλ/β Cos θ. Here, D is the average crystalline size (nm), k is the Scherrer constant with the value from 0.9 to 1, λ is the X-ray wavelength, β is the full width of half maximum, and θ is the Bragg diffraction angle (degrees). The estimations included stress investigation, remaining austenite quantitation, crystallite capacity, crystallinity consideration, and materials examination through overlaid X-ray diffraction models. Finally, Se-NPs concentration assessment was performed using UNICAM939 Atomic Absorption Spectroscopy, Cambridge, UK, and implemented with deuterium experience improvement. All suspensions were prepared using ultra-pure water [34,35,36,37,38,39,40,41]. Furthermore, the morphology size of the manufactured NPs was read by practicing TEM microscopy, JEOL JEM-100 CX, (Peabody, MA, USA).

### 2.3. Control of Rhizoctonia solani by Se-NPs

#### 2.3.1. Source of Pathogen and Culture Conditions

*Rhizoctonia solani* RCMB 031001 was purchased from the Regional Center for Mycology and Biotechnology (RCMB), Al-Azhar University, Cairo, Egypt. *R. solani* was cultured on potato dextrose agar medium (PDA) plates, incubated for 3–5 days at 28 ± 2 °C, and then kept at 4 °C for further use [42,43,44,45].

#### 2.3.2. In Vitro Assessment of Antifungal Activity and Growth Inhibition

Well diffusion method

The well diffusion method was applied to study the antifungal activity of biosynthesized Se-NPs [46] with a few modifications. *R. solani* was inoculated on PD broth medium and then incubated at 28 ± 2 °C for 3–5 days. Fungal inoculum of *R. solani* RCMB 031001 was spread thoroughly on the sterilized solidified potato dextrose agar (PDA) medium. At the same time, eight wells with 5.5 mm diameter were made using a sterile cork-borer on each agar plate (120 mm). The wells were filled with 50 µl of different concentrations of Se-NPs individually with triplicates. The culture plates were incubated at 25 °C for 7 days, and the zones of inhibition were observed and measured.

Radial growth method

PDA medium was prepared and amended with different concentrations of Se-NPs (1, 0.5, 0.25, 0.125, and 0.0625 mM) before the pouring stage. After medium solidification, culturing of *R. solani* was carried out according to Joshi et al. [47]. The inhibition percentage of pathogen growth was calculated using the following equation:$$\mathrm{Inhibition}\;\mathrm{of}\;\mathrm{pathogen}\;\mathrm{growth}\;\left(\%\right)=\frac{\mathrm{Growth}\;\mathrm{in}\;\mathrm{the}\;\mathrm{control}-\mathrm{Growth}\;\mathrm{in}\;\mathrm{the}\;\mathrm{treatment}}{\mathrm{Growth}\;\mathrm{in}\;\mathrm{the}\;\mathrm{control}}\times 100.$$

#### 2.3.3. In Vivo Assessment Efficacy of Se-NPs on *Vicia faba*

The inoculum of the pathogenic fungus *R. solani* was prepared according to Büttner et al. [48] that comprises mixing contents of 5 pure *R. solani* culture Petri dishes with 1000 mL of distilled water using electrical blender for two minutes. This experiment was carried out in the garden of Plant and Microbiology department, Faculty of Science, Al-Azhar University, Cairo, Egypt. The source of faba bean cv. Giza 716 was obtained from the Legume Research Department, Field Crop Institute, Agricultural Research Center, Egypt. The sandy loam soil was autoclaved (1.5 atm, 121 °C for 30 min) and distributed equally in disinfected pottery pots (30 cm in diameter) with 12 sterilized seeds per pot. The *Vicia faba* seeds were washed with distilled water then sterilized using 2% sodium hypochlorite for 2 min before conducting the treatments shown in Table 1.

#### 2.3.4. Disease Symptoms and Disease Index

Pre-emergence damping-off was measured after 15 days from sowing, while post-emergence damping-off and survival were measured after 30 days from sowing according to Mousa et al. [49]. In addition, disease symptoms were assessed, and the disease index was recorded after 45 days from sowing according to Grünwald et al. [50]. The disease index scale (0–5) based on disease progress developed by the authors was used to measure the disease severity of *Rhizoctonia* root rot, in which 0 indicated no visible symptoms; 1, a few small soft lesions on a part of the root system and hypocotyls; 2, elongated, discolored lesions spread on the entire root system and hypocotyls; 3, deep brown necrosis grind the stem, partial root disintegration, and yellowing of leaves; 4, stem canker, root disintegration, yellowing of leaves, and stunting; and 5, collapse and death of the plants. Disease index = (i (rating no. × no. of plants in the rating)/(total no. of plants × highest rating) × 100. Shoot length, root length, fresh and dry weight, and pigments were also measured (one gram of fresh leaves was extracted by 100 mL of 80% aqueous acetone (*v*/*v*), filtrated, and then completed the volume to 100 mL using 80% acetone). The optical density of the plant extract was measured using the spectrophotometer of three wavelengths (470, 649, and 665 nm). Pigments were calculated using the equations mentioned Mg chlorophyll (a)/g tissue = 11.63(A665) − 2.39(A649), Mg chlorophyll (b)/g tissue = 20.11(A649) − 5.18(A665), Mg chlorophyll (a + b)/g tissue = 6.45 (A665) + 17.72(A649), and Carotenoids = 1000 × O.D_470_ − 1.82 C_a_ − 85.02 C_b_/198 = mg/g fresh weight. “A” denotes the reading of optical density, phenol (one gram dry leaves was extracted with 80% cold methanol (*v*/*v*) three times at 0 °C. The extract was filtered; then, the volume of sample was completed to 25 mL with cold methanol. The total phenol and total soluble protein of plants were determined in the following manner: one gram of the dried leaves was added to 5 mL of 2% phenol water and 10 mL of distilled water was added; the solution was shaken and kept overnight, filtered, and completed volume to 50 mL with distilled water; then the protein content was determined according to Alhaithloul et al. [51].

### 2.4. Statistical Analysis

Experimental data were subjected to one-way analysis of variance (ANOVA) and the differences between means were measured using Tukey’s method. The values are given as means ± SD (standard deviations). Levels of significance were considered at *p* ≤ 0.05 unless otherwise stated, and the (L.S.D) at 5% level of probability using Co-state software [52].

## 3. Results and Discussion

### 3.1. Synthesis and Characterization of Se-NPs

In the current study, the supernatant of *Bacillus megaterium* ATCC 55000 was used to synthesize Se-NPs. The process of selenious acid reduction was monitored, while the cell-free extract changed from colorless to reddish color [53,54]. The UV-visible spectrum of Se-NPs synthesized indicated that it had maximum absorption at (0.860 abs) and 435 nm. DLS was performed to evaluate the particle size distribution, and the average particle size was found to be 45.9 nm, as shown in Figure 1B. On the other hand, the TEM result demonstrated that particles had a spherical shape within a nanoscale range from 29.72 to 74.36 nm with an average of the main diameter of 41.2 nm, as shown in Figure 1C. The XRD pattern for the Se-NPs was presented in Figure 1D. Several peaks were observed at nine theta (degree) as 23.2°, 30.5°, 41.7°, 44.3°, 46.4°, 52.3°, 56.7°, 62.5°, and 72.6° corresponding to the (100), (101), (110), (102), (111), (201), (113), (202), and (210) planes of the standard cubic phase of Se, respectively. The XRD pattern indicated that Se-NPs were in the face-centered cubic (FCC) structure and crystal in nature. The observation of diffraction peaks for the Se-NPs indicated that they were crystalline, while their refining was related to the particles in the nanometer size regime. The strong interaction of the Se-NPs with light was the result of the electrons conducting on the metal surface that were subjected to a collective oscillation when excited by light at specific wavelengths, which is known as surface plasma resonance (SPR) [55,56]. In another study, the culture supernatant of *A. terreus* with SeO_2_ (100 μg/mL) produced Se-NPs with an average size of 47 nm [57]. *Bacillus megaterium* (a halophile strain) strongly reduced selenite (up to 0.25 mM) to Se-NPs after 40 h of incubation [58]. A microbial source *Bacillus cereus-*mediated synthesis of Se-NPs showed an absorption maxima at 590 nm, whereas nanoparticles synthesized from lemon leaf extract exhibited a maximum absorption at 395 nm [59]. The band gap energy calculated for chemically formed nano-Se was 2.1 eV, which significantly different from a biological source (band gaps for nano-Se from *Sulphurospirillum barnessi, Bacillus selenitireducens,* and *Selenihalanaerobacter shriftii* were 1.62, 1.67, and 1.52 eV, respectively [60].

### 3.2. In Vitro Control of R. solani

#### 3.2.1. Antifungal Activity of Se-NPs and Minimum Inhibition Concentration

Metal nanoparticles such as silver nanoparticles [61,62], copper nanoparticles [63], and zinc nanoparticles [64] are wildly used for controlling fungal plant pathogens. However, selenium nanoparticles have strong antifungal activity, while they are rarely used for controlling fungal plant pathogens. Therefore, selenium nanoparticles were biosynthesized in this study to control *R. solani*. The antifungal activity of Se-NPs was assessed against *R. solani* using the agar well diffusion method; different concentrations of Se-NPs ranging from 1 to 0.0078 mM were tested as antifungal agent, as shown in Figure 2. Results illustrated that concentrations of Se-NPs of 1, 0.5, 0.25, 0.125, and 0.0625 mM had antifungal activity against *R. solani.* Moreover, 1 mM of Se-NPs had the maximum antifungal activity and gave an inhibition zone of 45 mm, whereas 0.0625 mM had the lowest antifungal activity against *R. solani* and gave an inhibition zone of 12 mm. From these data, 0.0625 was the minimum inhibition concentration for the controlling of *R. solani*.

#### 3.2.2. Effect of Se-NPs on Linear Growth of *R. solani* and Minimum Fungicidal Concentration

The linear growth of *R. solani* was assessed at different concentrations of Se-NPs with different incubation periods from 1 to 7 days, as shown in Figure 3A,C. Linear growth was performed to detect the inhibition percentage for each concentration of Se-NPs against *R. solani*. Results illustrated that the inhibition percentage increased with increasing of concentration Se-NPs, while linear growth decreased, as shown in Figure 3B. At concentration 1 mM, *R. solani* could not grow on a PDA surface, as shown in Figure 3C, inhibition percentage was 100%, and this concentration had the minimum fungicidal activity. Additionally, Se-NPs at 0.5 mM gave a high inhibition percentage but less than at 1 mM, where it was 92.9%; also, inhibition percentage decreased gradually with decreasing the concentration of Se-NPs [47]. Biosynthesized Se-NPs could suppress the growth and proliferation of *Sclerospora graminicola* [65]. Moreover, selenium nanoparticles were used in controlling the leaf blight of tomato caused by *Alternaria alternate,* and Se-NPs at a concentration of 100 ppm gave an inhibition percentage 89.6% [66]. In addition, Se-NPs used against *Alternaria solani* caused Early Blight Disease on Potato, and inhibition percentage was 100% at 800 ppm [67].

### 3.3. In Vivo Control of R. solani

#### 3.3.1. Efficacy of Se-NPs on Rhizoctonia Root Rot Disease of *Vicia faba* under Pot Conditions

The results presented in Table 2 and Figure 4 indicated that *R. solani* RCMB 031001 caused an emergence damping-off disease of 58.33% seeds and 88% Rhizoctonia root rot disease index of *Vicia faba* cultivar (treatment 2 infected control). On the other hand, the healthy control treatment pots resulted in 100% emerged and survived plants. These results confirmed that the cv. Giza 716 faba bean cultivar is susceptible to *R. solani* RCMB 031001. Application of the Se-NPs by soaking and/or spraying to *Vicia faba* infested with fungus *R. solani* showed greater potency in controlling the pathogen. The best treatment for controlling *R. solani* was treatment 6, which resulted in 83.33% survival as well as 72.27% protection followed by treatment 4 with 66.67% and 59.2%, respectively, and treatment 8 by 50% and 63.63%. These results are similar to studies by Nandini, Hariprasad, Prakash, Shetty and Geetha [65], which reported that Se-NPs had a highly effective role in controlling plant pathogenic fungus *Sclerospora graminicola* as a causative agent of downy mildew disease. However, several reports showed that the application of selenium in plants activates the defense plant mechanism against abiotic [68] and biotic stresses such as *R. solani* [69]. 

#### 3.3.2. Growth and Yield Responses of *Vicia faba* by Se-NPs under Pot Conditions

Results presented in Table 3 and Table 4 indicated that all investigated growth parameters (shoot and root length, number of leaves, fresh and dry weight plant biomass), as well as yield (number of bods per plant, number of seeds per plant, the weight of 100 seed and protein content of yield) of infected *Vicia faba* cv. Giza 716 plants with *R. solani* RCMB 031001 were significantly decreased compared with healthy control plants. The most effective treatment was treatment 6, which increased the yield and growth parameters, especially shoot dry weight 203%, root fresh weight 178.8%, root dry weight 163%, plant height, and number of seeds 116.6% compared with infected control (treatment 2). These results are similar to Abdel-Monaim [70], who reported that *R. solani* had significantly decreased fresh and dry weight compared to healthy control. Akladious et al. [71] reported that *R. solani* caused a significant decrease in shoot length, root length, number of leaves, and fresh and dry weight of shoot and root of *Vicia faba*.

The obtained results revealed that all investigated growth parameters of *Vicia faba* cv. Giza 716 plants were significantly increased in response to the application of Se-NPs compared with the control. The simulative effects of Se-NPs on plant growth were explained by many mechanisms—firstly, the increased starch content in chloroplast [72]. Secondly, the plant cell can be protected by selenium from oxidative damage by antioxidant defenses [73]. Thirdly, selenium is a beneficial element for plants and has a bio-stimulant effect, as photocatalysis and plant growth increase plant metabolism and crop quality and stress tolerance [32,52,74]. However, the application of selenium in plants stimulates the growth and quality of fruits [75]. Our results showed that the most effective treatment was achieved by soaking and foliar spray followed by soaking and finally spraying.

#### 3.3.3. Effect of Se-NPs on Photosynthetic Pigments of *Vicia faba* under Pot Conditions

The observed results in Figure 5 showed that chlorophyll content and carotenoids had significantly decreased by *R. solani* RCMB 031001. These results are explained with [76], which stated that phytopathogenic fungi inhibit the photosynthetic activity of plants. These reductions in chlorophyll a may be due to the more selective destruction of chlorophyll biosynthesis or degradation of chlorophyll precursors according to Saha et al. [77] or may be due to a decrease in the uptake of minerals (e.g., magnesium) that are required for chlorophyll synthesis and interfere with the photosynthesis reactions [78]. Data presented in Figure 5 indicated that the application of Se-NPs caused a significant increase in total chlorophyll content and carotenoids compared with controlled plants and the best result was achieved by soaking and foliar spraying. Several reports show that the application of selenium in plants improves photosynthesis [79].

#### 3.3.4. Effect of Se-NPs on the Metabolic Indicators of (*Vicia faba* L.)

##### Effect of Se-NPs on Phenol Contents of Vicia faba under Pot Conditions

Results revealed that the contents of total phenols were significantly increased in shoots and roots of cv. Giza 716 plants in response to the infection with *R. solani* RCMB 031001, as shown in Figure 6. Moreover, results demonstrated that application of Se-NPs induced responses regarding the total contents of phenols compared with healthy control. In contrast, total phenols contents in shoots and roots-infected plants were significantly decreased in response to the treatments with Se-NPs. These results are similar to those in [80,81]; they demonstrated that the treatment of plants with NPs resulted in increasing phenolic content. This increasing in phenolic contents resulted in antifungal activity by several mechanisms including (i) cell rupture and release of intracellular proteins and carbohydrates that prevent fungal growth; (ii) inhibition of mitochondrial respiration causing reduction of ATP production, and (iii) oxidative lesions and chelation of iron ions [82,83]. Correspondingly, total phenols play a vital role in the regulation of plant metabolic process and overall plant growth as well as lignin synthesis [84]. Phenols act as free radical scavengers as well as substrates for many antioxidant enzymes [85]. Finally, Mellersh et al. [86] reported that reactive oxygen species (ROS), especially phenolic compounds, prevent penetration, restrict fungal growth, and provoke cell death and tissue necrosis, which would prevent further fungal development toward plant tissue.

##### Effect of Se-NPs on a Total Soluble Protein of *Vicia faba* under Pot Conditions

The presented data in Table 5 showed that the total soluble protein in shoot and root were significantly decreased in cv. Giza 716 plants in response to the infection with *R. solani* RCMB 031001. Weintraub and Jones [87] recorded that pathogen attack resulted in a reduction of several thylakoid membrane proteins and decreasing leaf soluble protein. These results are explained by several different mechanisms; firstly, the stresses may affect the process of protein synthesis, Secondly, it is also possible that the pathogens consume nitrogen, which could have been utilized for synthesizing proteins [88]. In addition, the application of Se-NPs resulted in an increase of total soluble protein compared with control. Also, the best treatment was soaking and foliar spraying, which agree with Hajiboland [89], who illustrated that the application of Se-NPs resulted in a significant increase in total soluble protein. Increasing protein content could be due to the activation of the host defense mechanisms as an indicator of resistance [88].

##### Effect of Se-NPs on Oxidative Enzymes of *Vicia faba* under Pot Conditions

*Vicia faba* showed variation in relative mobility and density polypeptide bands as pathogenicity indicators and or treatment with Se-NPs, where healthy control (treatment 1) gave three isozyme bands with a low density of isozymes but soaking infected (treatment 4) and soaking *+* foliar spray Se-NPs (treatment 6) gave the same number of bands, three isozymes with moderate density. While infected control (treatment 2), as well as Se-NPs as a foliar spray on infected (treatment 8), gave three isozymes bands with a high density of isozymes, as shown in Figure 7, these results demonstrated that infected control recorded high activity as a high density of bands. Our results are similar to those of [90], who reported that the minimum activities of peroxidase enzymes were observed in healthy control. In this regard, Hasanuzzaman and Fujita [91] found that spraying with selenium increased the activity of many enzymes. In addition, [92] reported that nano selenium acts as a promoter and/or stressor, enhancing the antioxidant defense systems of plants, which leads to the improvement of plant tolerance under sandy soil conditions.

In addition to the lowest polyphenol oxidase (PPO) activity recorded in healthy control, Se nanoparticles and *R. solani* infection application showed variation in number, relative mobility, and density polypeptide bands more than healthy ones, infected control, as well as treatment with Se-NPs as a foliar spray on infected plants (four isozyme bands) with a high density of isozymes (Figure 8). Meanwhile, plants treated with soaking or foliar spray Se-NPs (either individual or combination) gave the same number of bands (four isozymes) with moderate and high density. Variation in isozyme shows knowledge of resistant genes in the biological system to physiological changes, genetic traits, and growth in various species _ENREF_ [90]. Finally, anti-oxidative enzymes such as polyphenol oxidase (PPO) and peroxidase (POX) are most importantly involved in the scavenging system of excess reactive oxygen species (ROS) [90].

## 4. Conclusions

In the current study, Se-NPs were bio-synthesized by the culture supernatant of *B. megaterium* ATCC 55000, which was characterized by mono-dispersed spheres with a mean diameter of 41.2 nm. The green Se-NPs have promising antifungal activity against *R. solani* in vitro and in vivo; hence, it could use as a promising agent for the controlling of *R. solani* diseases in faba bean. Se-NPs effects on faba bean plant growth and development at the working concentration were determined. *Vicia faba* plant growth promoters in Se-NPs were the enhancement of *Vicia faba*’s morphological, metabolic and genetic parameters. Photosynthetic pigments, metabolic indicators, and phenolics compounds of *Vicia faba* were analyzed; Se-NPs caused a significant increase in total chlorophyll content and carotenoids compared with controlled plants, and the best results were achieved by soaking and foliar spraying. Moreover, results demonstrated that the application of Se-NPs induced responses regarding the total contents of phenols and total soluble protein compared with healthy control. In contrast, total phenols contents in shoots and roots-infected plants were significantly decreased in response to the treatments with Se-NPs. The effects of Se-NPs on oxidative enzymes such as polyphenol oxidase (PPO) and peroxidase (POX) in *Vicia faba* under pot conditions were assayed. Se-NPs act as a promoter and/or stressor, enhancing the antioxidant defense systems of plants, which leads to the improvement of plant tolerance. It is widely demanded that biogenic selenium NPs may be effective and economical alternatives for treating fungal plant pathogens. In the future, the adverse effects of these biogenic NPs on agriculture and ecosystems should be ascertained before their commercial use in plant disease control in the field.

## Figures and Tables

**Figure 1 jof-07-00195-f001:**
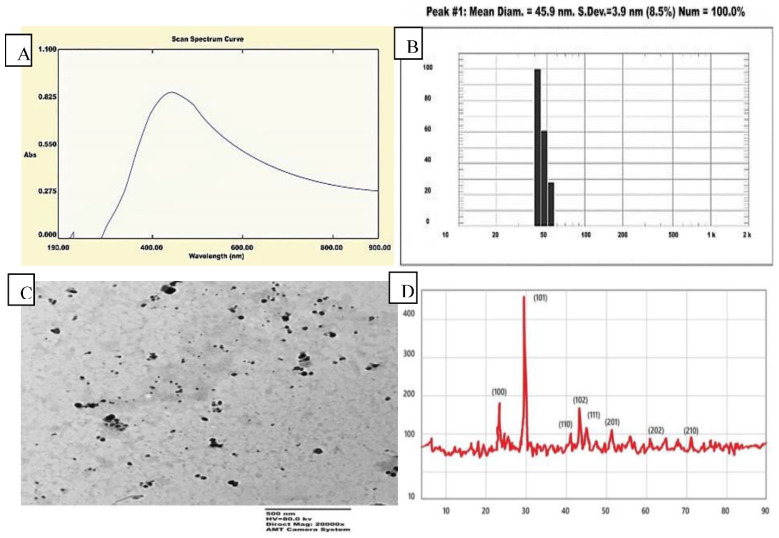
Characterization of bacteriogenic Se-NPs produced by *B. megaterium* (**A**–**D**); (**A**) UV-Visible spectrum; (**B**) dynamic light scattering (DLS); (**C**) TEM image; (**D**) XRD.

**Figure 2 jof-07-00195-f002:**
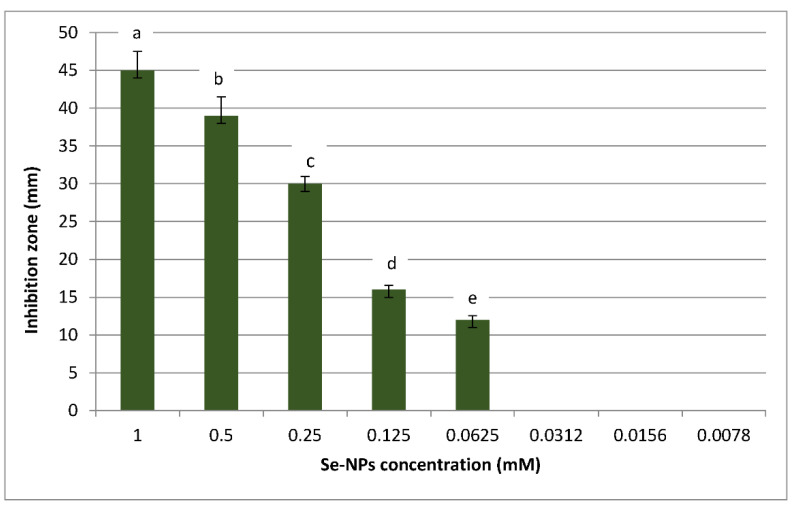
Antifungal activity of different concentrations of Se-NPs against *R. solani.* Data are expressed as means ± standard deviations of triplicate assays. The different alphabetic superscripts in the same column are significantly different (*p* < 0.05) based on Tukey’s multiple comparison test.

**Figure 3 jof-07-00195-f003:**
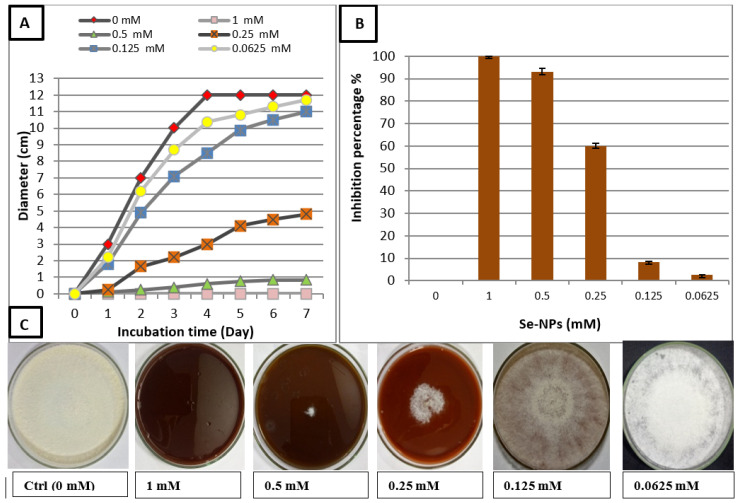
Effect of Se-NPs on *R. solani* (**A**–**C**): (**A**) Linear growth at different incubation periods from 0 to 7 days; (**B**) Inhibition percentages of *R. solani* at different concentration of Se-NPs; (**C**) Linear growth on potato dextrose agar medium (PDA) plates at 7 days. Data are expressed as means ± standard deviations of triplicate assays. The different alphabetic superscripts in the same column are significantly different (*p* < 0.05) based on Tukey’s multiple comparison test.

**Figure 4 jof-07-00195-f004:**
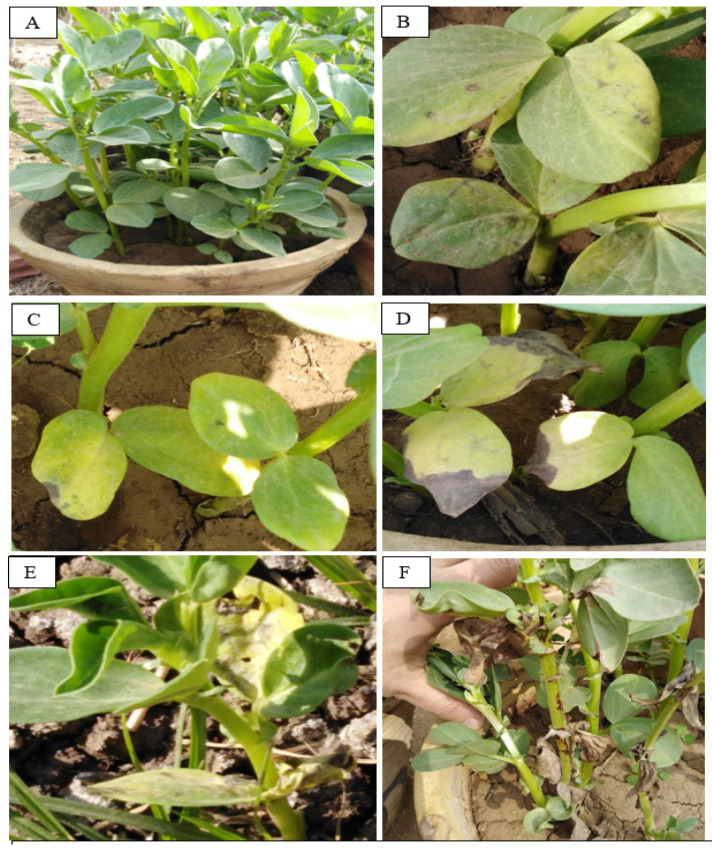
Disease index scale (0–5); (**A**): 0, (**B**): 1, (**C**): 2, (**D**): 3, (**E**): 4, and (**F**): 5; the disease index was recorded after 45 days from sowing.

**Figure 5 jof-07-00195-f005:**
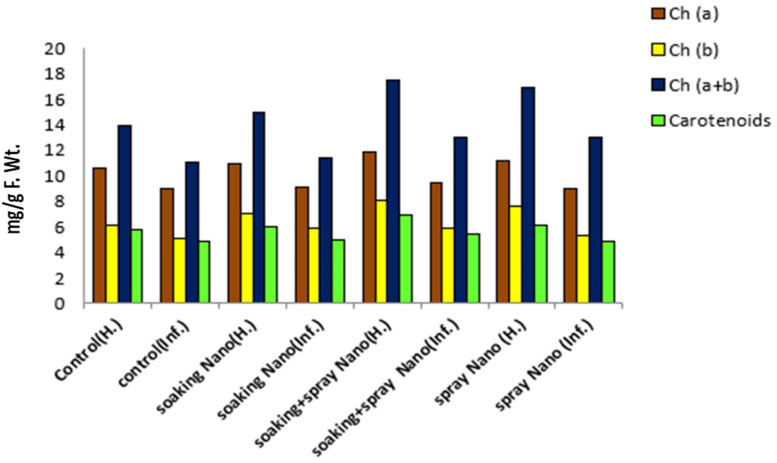
Effect of Se-NPs on the photosynthetic pigment’s indicators of (*Vicia faba* L.).

**Figure 6 jof-07-00195-f006:**
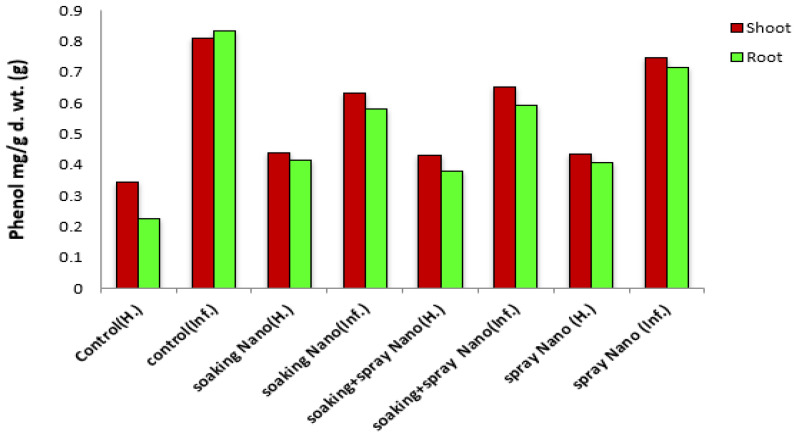
Effect of Se-NPs on the phenolics compounds of (*Vicia faba* L.).

**Figure 7 jof-07-00195-f007:**
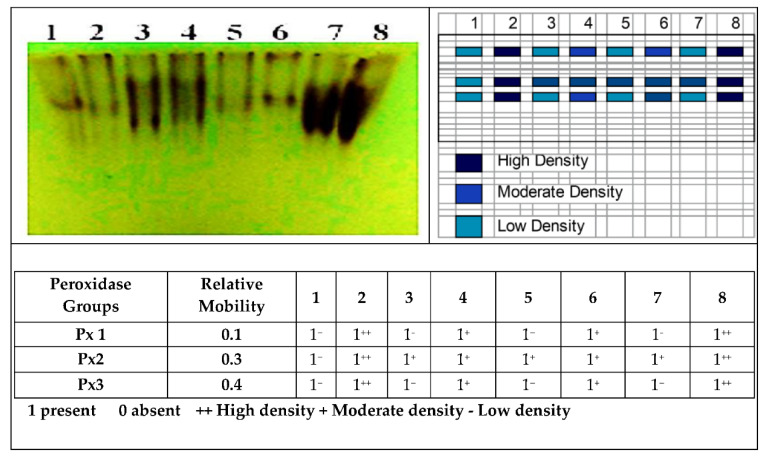
Effect of Se-NPs on peroxidase isozymes of *Vicia faba* under pot conditions 1: Control (H.). 2: Control (Inf.). 3: Soaking Nano (H.). 4: Soaking Nano (Inf.). 5: Soaking + Spray Nano (H.). 6: Soaking + Spray Nano (Inf.). 7: Spray Nano (H.). 8: Spray Nano (Inf.).

**Figure 8 jof-07-00195-f008:**
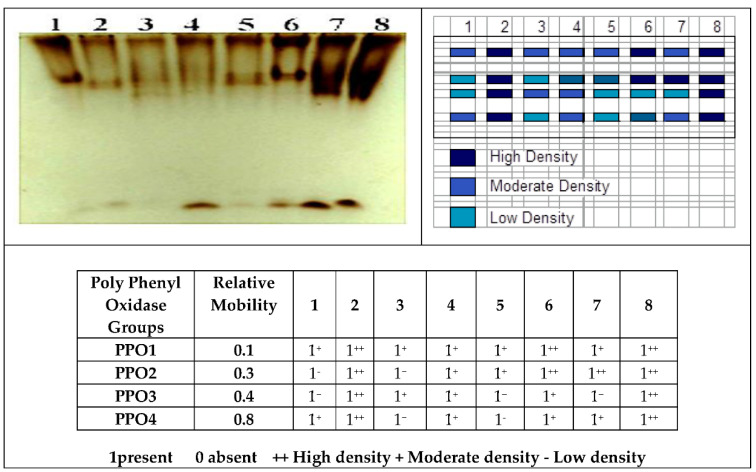
Effect of Se-NPs on Polyphenol oxidase isozymes of *Vicia faba* under pot conditions Control (H.). 2: Control (Inf.). 3: Soaking Nano (H.). 4: Soaking Nano (Inf.). 5: Soaking + Spray Nano (H.). 6: Soaking + Spray Nano (Inf.). 7: Spray Nano (H.). 8: Spray Nano (Inf.).

**Table 1 jof-07-00195-t001:** Treatments used in this study.

Treatment Number	Treatment
1 (Control healthy)	The sterilized *Vicia faba* seeds submerged in distilled water for three hours and sowing in sterilized soil.
2 (Control infected)	Sowing the sterilized *Vicia faba* seeds in distilled water for three hours and sowing in inoculated soil with *R. solani*.
3 (Healthy + Nano soaking)	Soaking the sterilized *Vicia faba* seeds in Se-NPs (0.0625 mM) for three hours and sowing in sterilized soil.
4 (Infected + soaking)	Soaking the sterilized *Vicia faba* seeds in Se-NPs (0.0625 mM) for three hours and sowing in inoculated soil with *R. solani*.
5 (Healthy + soaking and spraying with Nano)	Soaking the sterilized *Vicia faba* seeds in Se-NPs (0.0625 mM) for three hours and sowing in sterilized soil, then spraying 15 mL of Se-NPs after emergence.
6 (Infected + soaking and spraying with Nano)	Soaking the sterilized *Vicia faba* seeds in Se-NPs (0.0625 mM) for three hours and sowing in inoculated soil with *R. solani*, then spraying 15 mL of Se-NPs after emergence.
7 (Healthy + spraying Nano)	Sowing the sterilized *Vicia faba* seeds in distilled water for three hours and sowing in sterilized soil, then spraying 15 mL of Se-NPs (0.0625 mM) after emergence.
8 (Infected + spraying Nano)	Sowing the sterilized *Vicia faba* seeds in distilled water for three hours and sowing in inoculated soil with *R. solani*, then spraying 15 mL of Se-NPs (0.0625 mM) after emergence.

**Table 2 jof-07-00195-t002:** Effect of selenium nanoparticles (Se-NPs) on the disease index of *R. solani* damping-off and root rot diseases under pot conditions.

Treatment		Pre-Emergence Damping of %	Post-Emergence Damping of %	Survival Plant %	Disease Index %	Protection %
**Healthy**	Control	0	0	100	0	-
	Nano soaking	0	0	100	0	-
	Nano spraying	0	0	100	0	-
	Nano (soaking + spraying)	0	0	100	0	-
**Infected**	Control	50	8.33	41.67	88	0
	Nano soaking	16.66	16.66	66.67	36	59.2
	Nano spraying	50	0	50	32	63.63
	Nano (soaking + spraying)	16.667	0	83.33	20	77.27

**Table 3 jof-07-00195-t003:** Effect of biogenic Se-NPs on morphological indicators of *Vicia faba L*. under pot conditions.

Treatments	Plant Height (cm)	Root Length (cm)	Number of Leaves	Shoot F. wt. (g)	Shoot D. wt. (g)	Root F. wt. (g)	Root F. wt. (g)
T1: Control (H.)	32 ± 1.50 ^b^	11.33 ± 0.85 ^cd^	16.33 ± 0.57 ^bc^	10.48 ± 0.72 ^bc^	3.71 ± 0.24 ^b^	1.25 ± 0.04 ^d^	0.37 ± 0.00 ^bc^
T2: Control (Inf.)	19.36 ± 0.70 ^e^	9.73 ± 0.75 ^d^	11.66 ± 0.57 ^d^	7.38 ± 0.33 ^e^	1.31 ± 0.28 ^d^	0.82 ± 0.06 ^g^	0.19 ± 0.02 ^d^
T3: Soaking Nano (H.)	35.83 ± 1.89 ^b^	13.56 ± 0.51 ^ab^	18.33 ± 1.52 ^b^	13.56 ± 0.40 ^a^	4.6 ± 0.35 ^a^	1.64 ± 0.06 ^b^	0.52 ± 0.04 ^a^
T4: Soaking Nano (Inf.)	20.5 ± 1.80 ^de^	10.7 ± 0.75 ^cd^	15.33 ± 1.15 ^c^	10.25 ± 0.67 ^bc^	2.48 ± 0.14 ^c^	1.07 ± 0.06 ^ef^	0.22 ± 0.00 ^d^
T5: Soaking + Spray Nano (H.)	42.66 ± 2.25 ^a^	15.66 ± 1.10 ^a^	22.33 ± 1.52 ^a^	14.76 ± 0.92 ^a^	4.59 ± 0.22 ^a^	1.87± 0.03 ^a^	0.57 ± 0.04 ^a^
T6: Soaking + Spray Nano (Inf.)	24.5 ± 0.50 ^cd^	10.84 ± 0.74 ^cd^	11.66 ± 0.57 ^d^	8.37± 0.10 ^de^	2.67 ± 0.30 ^c^	0.93 ± 0.06 ^fg^	0.31 ± 0.00 ^c^
T7: Spray Nano (H.)	34.5 ± 1.32 ^b^	12.06 ± 0.62 ^bc^	16.33 ± 0.57 ^bc^	11.16 ± 0.35 ^b^	3.92 ± 0.13 ^ab^	1.45 ± 0.09 ^c^	0.44 ± 0.00 ^b^
T8: Spray Nano (Inf.)	26.66 ± 1.52 ^c^	10.6 ± 0.65 ^cd^	13.66 ± 0.57 ^cd^	9.46 ± 0.47 ^cd^	2.65 ± 0.16 ^c^	1.13 ± 0.01 ^de^	0.35 ± 0.00 ^c^
L.S.D at 0.05	2.667	1.329	1.694	0.963	0.422	0.099	0.041

H. means Healthy and Inf. means infected. Data are expressed as means ± standard deviations of triplicate assays. The different alphabetic superscripts in the same column are significantly different (*p* < 0.05) based on Tukey’s multiple comparison test.

**Table 4 jof-07-00195-t004:** Effect of Se-NPs on the yield of (*Vicia faba* L.) plants.

Treatments	No. of Pods/Plant	No. of Seeds/Plant	wt. of 100 Seeds(g)	Protein Yield mg/g (g)
T1: Control (H.)	20.33 ± 0.57 ^a^	51 ± 1.0 ^bc^	101.33 ± 0.57 ^c^	116.94 ± 0.09 ^c^
T2: Control (Inf.)	17.66 ± 0.57 ^b^	47 ± 2.0 ^c^	99 ± 1.0 ^d^	96.1 ± 0.28 ^f^
T3: Soaking Nano (H.)	21.33 ± 0.57 ^a^	54.66 ± 1.52 ^ab^	105 ± 1.0 ^b^	120.6 ± 0.46 ^b^
T4: Soaking Nano (Inf.)	18.33 ± 0.57 ^b^	49 ± 1.73 ^c^	101.66 ± 0.57 ^c^	99.02 ± 0.14 ^e^
T5: Soaking + Spray Nano (H.)	21.66 ± 0.57 ^a^	57.66 ± 1.15 ^a^	107.03 ± 0.45 ^a^	123.21 ± 0.38 ^a^
T6: Soaking + Spray Nano (Inf.)	17.33 ± 0.57 ^b^	49 ± 1.73 ^c^	101.16 ± 0.28 ^c^	101.28 ± 0.51 ^d^
T7: Spray Nano (H.)	20.33 ± 0.57 ^a^	54.33 ± 1.15 ^ab^	103.5 ± 0.05 ^b^	118.17 ± 0.32 ^c^
T8: Spray Nano (Inf.)	18 ± 1.0 ^b^	51 ± 1.0 ^bc^	101.30 ± 0.02 ^c^	100.21 ± 1.05 ^de^
L.S.D at 0.05	1.125	2.523	1.095	0.856

Data are expressed as means ± standard deviations of triplicate assays. The different alphabetic superscripts in the same column are significantly different (*p* < 0.05) based on Tukey’s multiple comparison test. LSD (*p* < 0.05) values are indicated in the data differing significantly are indicated with different letters.

**Table 5 jof-07-00195-t005:** Effect of Se-NPs on the total soluble protein of (*Vicia faba* L.).

Treatments	Protein Shoot mg/g d. wt. (g)	Protein Root mg/g d. wt. (g)
T1: Control (H.)	20.22 ± 0.15 ^c^	18.27 ± 0.17 ^c^
T2: Control (Inf.)	16.8 ± 0.24 ^f^	16.63 ± 0.35 ^f^
T3: Soaking Nano (H.)	21.83 ± 0.06 ^b^	19.14 ± 0.17 ^b^
T4: Soaking Nano (Inf.)	18.06 ± 0.48 ^e^	17.39 ± 0.07 ^e^
T5: Soaking + Spray Nano (H.)	22.93 ± 0.05 ^a^	21.03 ± 0.07 ^a^
T6: Soaking+ Spray Nano (Inf.)	19.09 ± 0.08 ^d^	17.59 ± 0.07 ^de^
T7: Spray Nano (H.)	21.62 ± 0.04 ^b^	20.9 ± 0.09 ^a^
T8: Spray Nano (Inf.)	18.08 ± 0.31 ^e^	18 ± 0.17 ^cd^
LSD at 0.05	0.412	0.283

Data are expressed as means ± standard deviations of triplicate assays. The different alphabetic superscripts in the same column are significantly different (*p* < 0.05) based on Tukey’s multiple comparison test.

## Data Availability

Not applicable.

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
