# Peer review of "Bacillus megaterium*-Mediated Synthesis of Selenium Nanoparticles and Their Antifungal Activity against *Rhizoctonia solani* in Faba Bean Plants"

_jof, 2021, doi:10.3390/jof7030195_

Round 1
Reviewer 1 Report
The article Bacillus-mediated synthesis of selenium nanoparticles and their antifungal activity against Rhizoctonia solani in Vicia faba presents data on the use of selenium bionanoparticles to improve plant growth Vicia faba and to protect faba bean against the phytopathogenic fungus Rhizoctonia solani.
However, the article was made extremely carelessly and needs serious revision. There are some methodological aspects that are not sufficiently described. Thus, the section Materials and Methods do not describe how the authors obtained the Se-NPs. It seems that they fell from the sky. To obtain Se-NPs, at first, it is necessary to carry out isolation and purification. However, there is no such description in the presented manuscript. And such a description is also missing in the links mentioned by the authors (references 22-24). We recommend the authors to familiarize with the article: A.V. Tugarova, P.V. Mamchenkova, Yu.A. Dyatlova, A.A. Kamnev FTIR and Raman spectroscopic studies of selenium nanoparticles synthesised by the bacterium Azospirillum thiophilum Spectrochim. Acta Part A: Mol. Biomol. Spectrosc. 2018 (this article is in attachment). Various carbohydrates and proteins (peptides) are associated with the surface of nanoparticles during the release of biosynthesized nanoparticles from bacteria and other microorganisms. Therefore, these associated molecules can also contribute to the various biological effects of nanoparticles. Hence, it is difficult to separate the effect of the nanoparticles themselves from the effect of associated molecules. In addition, in the results presented by the authors, there is no control treatment with elemental selenium (molecules), which does not form relatively large particles in comparison with free molecules. What could be the reason for the bioeffect: with nanoparticles themselves or with molecules that dissociate from the surface of the particles? What if the treatment were only elemental selenium without nanoparticles?
In the Introduction and Discussion, references are made to little-known works published in journals with low impact, it is desirable to correct this situation.
The manuscript contains a lot of inaccuracies, misprints, repetitions, and English correction is required.
Specific comments are listed below.
Line 3 - species names according to the international nomenclature - Vicia faba L. Latin names should preferably be given in italics.
Line 5 - the letter in the last name of the last author
Line 43 - extra square brackets
Line 56 - Solani – solani
Line 152 - you need to put the designation: 2.3.4. Disease symptoms and disease index
Line 194 - Figure 1 - you need to improve the sharpness of the image, letters and numbers are difficult to see
Line 205 no punctuation mark
Line 232 - The drawing style is different from the previous one. The quality of the images on the graph requires a sharpness correction on the histogram. There is no information on the statistical characteristics of the experiment and errors
Line 255 - There is no accepted scoring description of damage, and there is no clear description of images, age, time of infection, etc.
Line 278 - Below Table 2 there is no data on statistical processing, the edge of the table is not decorated, in the first column in the fifth line s - uppercase, apparently, a capital one is needed
Lines 278, 280 - Two tables side by side and decorated in different ways, you can't! This is untidy. Correct uppercase letters to capital letters in Table 3 on lines 5, 7, 9
Lines 283, 287, 289 - remove the bold in the text
Line 293 - two identical links
Line 294 - no statistical error
Line 311, 313, 319, 321,328, 341, 349, 353, 358 - change bold
Line 320 - Is Giza716 a breeding line or a cultivar?
Line 331 - no statistical description below the table
Line 356 - improve the sharpness of the figure and captions.

Author Response
Thank you very much for reviewing our manuscript and giving us helpful comments and valuable recommendations. Many thanks for your agreement, and we hope that the revised manuscript meets your approval and full satisfaction. We revised the paper according to your specific comments. Detailed explanations for the comments are shown below.
|
1- There are some methodological aspects that are not sufficiently described. Thus, the section Materials and Methods do not describe how the authors obtained the Se-NPs. It seems that they fell from the sky. To obtain Se-NPs, at first, it is necessary to carry out isolation and purification. However, there is no such description in the presented manuscript. And such a description is also missing in the links mentioned by the authors (references 22-24). Response SeNPs nanoparticles were synthesized using standard bacterial strain (Bacillus megaterium ATCC 55000), So we not need for necessary to carry out isolation and purification. Also, we mention method of synthesis using B. megaterium supernatant. "SeNPs were manufactured using Bacillus megaterium culture supernatant. (as reducing and stabilizing agents). The selenious acid suspension (1mM) was mixed with cell free supernatant by quotient (1:1) v/v. The mixtures were stirred at controlled room temperature of around 25°C. The absorption color of the individual specimen was reported practicing a JASCO V-560 UV-visible spectrometer performing on a resolution at one nm |
|
2- We recommend the authors to familiarize with the article: A.V. Tugarova, P.V. Mamchenkova, Yu.A. Dyatlova, A.A. Kamnev FTIR and Raman spectroscopic studies of selenium nanoparticles synthesised by the bacterium Azospirillum thiophilum Spectrochim. Acta Part A: Mol. Biomol. Spectrosc. 2018 (this article is in attachment). Response It was taken in considered and added to our bibliography |
|
3- Various carbohydrates and proteins (peptides) are associated with the surface of nanoparticles during the release of biosynthesized nanoparticles from bacteria and other microorganisms. Therefore, these associated molecules can also contribute to the various biological effects of nanoparticles. Hence, it is difficult to separate the effect of the nanoparticles themselves from the effect of associated molecules. Response We agree with this concept, due to the active metabolites which secreted extracellular the bacterial cells such as carbohydrates and proteins (peptides) act as reducing agent to reduce the metal into nano size and some of their still associated out surface of nanoparticles and play as capping agent with increasing the activity of nanoparticles , however this considered advantage for biosynthesis of nanoparticles. |
|
4- The results presented by the authors, there is no control treatment with elemental selenium (molecules), which does not form relatively large particles in comparison with free molecules. What could be the reason for the bioeffect: with nanoparticles themselves or with molecules that dissociate from the surface of the particles? What if the treatment were only elemental selenium without nanoparticles? Response The aim of the work is to obtain a product that has a great effectiveness against Rhizoctonia solani, as well as an effectiveness in stimulating the growth of the Vicia faba plant as an alternative to chemical pesticides, and from previous research we find that nanomaterials give greater efficiency. The nanoparticles of Se (SeNPs) possess a lowest toxicity in compared to Selenium, that is why we used selenium in nanoform rather in element form. |
|
5- In the Introduction and Discussion, references are made to little-known works published in journals with low impact, it is desirable to correct this situation. Response Recent references were added in the revised manuscript |
|
6- The manuscript contains a lot of inaccuracies, misprints, repetitions, and English correction is required. Response Thank you very much for your comment. As you recommended we have carefully revised the manuscript and corrected grammatical and typos mistakes |
|
7- Line 3 - species names according to the international nomenclature - Vicia faba L. Latin names should preferably be given in italics. Response Modified |
|
8- Line 5 - the letter in the last name of the last author Response Corrected |
|
9- Line 43 - extra square brackets Response Amended |
|
10- Line 56 - Solani – solani Response Corrected |
|
11- Line 152 - you need to put the designation: 2.3.4. Disease symptoms and disease index Response Done |
|
12- Line 194 - Figure 1 - you need to improve the sharpness of the image, letters and numbers are difficult to see Response Done |
|
13- Line 205 no punctuation mark Response Corrected |
|
14- Line 232 - The drawing style is different from the previous one. The quality of the images on the graph requires a sharpness correction on the histogram. There is no information on the statistical characteristics of the experiment and errors Response Done |
|
15- Line 255 - There is no accepted scoring description of damage, and there is no clear description of images, age, time of infection, etc. Response It was explained in detail on lines 158, 159, 160, 161 and 162, and the age of the plant was added to the figure and images were changed |
|
16- Line 278 - Below Table 2 there is no data on statistical processing, the edge of the table is not decorated, in the first column in the fifth line s - uppercase, apparently, a capital one is needed Response Modified |
|
17- Lines 278, 280 - Two tables side by side and decorated in different ways, you can't! This is untidy. Correct uppercase letters to capital letters in Table 3 on lines 5, 7, 9 Response Modified |
|
18- Lines 283, 287, 289 - remove the bold in the text Response Removed |
|
19- Line 293 - two identical links Response Corrected |
|
20- Line 311, 313, 319, 321,328, 341, 349, 353, 358 - change bold Response Changed |
|
21- Line 320 - Is Giza716 a breeding line or a cultivar? Response Cultivar |
|
22- Line 331 - no statistical description below the table Response Statistical description was added |
|
23- Line 356 - improve the sharpness of the figure and captions Response Done |
|
24- How was peroxide mobility or density calculated or measure Response The obtained binary data were subjected to analysis with GelAnalyzer3 (Egygene) software. (Alzohairy, 2008). The similarity matrices were done using Gel works ID advanced software UVP- England Program.The relationships among genotypes and species as revealed by dendrograms were done by using SPSS windows (Version 16) program. Dice computerpackage was used to calculate the pairwise difference matrix and plot the phenogram among control and treatments genotype under investigation |

Reviewer 2 Report
The authors present a study on the utilization of selen nanoparticles for control of Rhizoctonia solani.
Although the claim biosynthesis they do not describe how synthesis was done, but they deliver a characterization of the particles. Unfortunately, they do not report on the importance of particle size. They also give no evidence that it is selen in their particles. Because synthesis is not described reader must believe it. For the outcome of the plant trials this is not crucial though. lets take it as a fact that they authors have synthesized selen nanoparticles. In the revison they should provide more evidences though.
The in vitro and in vivo trials are promising showing an effetcive control of Rhizoctonia. Especially a double treatment of soaking and spraying was effetcive. The authors do not mention if soaking for 3 days is common practice. Even for seed treatment this duration seems rather long. The authors should provide more information why they have chosen this duration and if it is applicable in practice.
The effects on plant growth are describe in a too short way. They authors do not highlight which result is important and why.
The same is true for the physiological parameter. Moreover, for these measurements no methods are given.
Major flaws of the manuscript.
(1) the language must be improved drastically. Some of the sentences are misleading and often not to understand. Only after reading sentences several times the reader could get the idea of the authors. I have commented on it on a very few ocasions only though (see PDF). The whole text must be revised completely and I strongly suggest to let it proof read by a native speaker.
(2) The tables and figures lack scientific standards. See my comments in the PDF file. The visualition is not suited for a scientif journal. All tables and figures MUST be revised completely. I would suggest to create new ones.
(3) Methods description is lacking important information (see comments in PDF). Some methods are note described at all.
(4) Data. For some data at least clarification and re-calculation is needed. Figures 7 and 8 are almost not to understand, because methods are missing and the descriptions with the fgraphs are not self-explaining. The authors do not deliver statistical evaluation for all results, but from the data given it is a must. On statistics: I strongly revommend to use the TUKEY-HSD instead of Duncan's multiple Range test. This test will give fals positive differences due to low number of replicates.
Impression:
I think the study could give important information for the scientific community. If the study would also be of importance for practice remains open, because the authors do not comment on it. The research design seems to be suited but description is lacking scientific standard.
To sum it up, an intersting study lacking scientific standards in description and presentation. The manuscript must be revised completely before it can be submitted and reviewed again. I would suggest to re-write it completely.

Author Response
Reviewer #2:
Thank you very much for reviewing our manuscript and giving us helpful comments and valuable recommendations. Many thanks for your agreement, and we hope that the revised manuscript meets your approval and full satisfaction. We revised the paper according to your specific comments. Detailed explanations for the comments are shown below and others are corrected in the revised manuscript through tracking option
|
1- Although the claim biosynthesis they do not describe how synthesis was done, but they deliver a characterization of the particles. Unfortunately, they do not report on the importance of particle size. They also give no evidence that it is selen in their particles. Because synthesis is not described reader must believe it. For the outcome of the plant trials this is not crucial though. lets take it as a fact that they authors have synthesized selen nanoparticles. In the revison they should provide more evidences though. Response "SeNPs were manufactured using Bacillus megaterium culture supernatant. (as reducing and stabilizing agents). The selenious acid suspension (1mM) was mixed with cell free supernatant by quotient (1:1) v/v. The mixtures were stirred at controlled room temperature of around 25°C. The absorption color of the individual specimen was reported practicing a JASCO V-560 UV-visible spectrometer performing on a resolution at one nm". Also, process of SeNPs synthesis depend on the reduction reaction which carry out after adding the selenious acid suspension (1mM) was mixed with cell free supernatant of B. megaterium. The reduction process associated with changing in colour to reddish brown and noted this change which indicated for SeNPs synthesis and then start in characterization step. |
|
2- The in vitro and in vivo trials are promising showing an effective control of Rhizoctonia. Especially a double treatment of soaking and spraying was effective. The authors do not mention if soaking for 3 days is common practice. Even for seed treatment this duration seems rather long. The authors should provide more information why they have chosen this duration and if it is applicable in practice. Response This part was corrected in the revised manuscript |
|
3- The effects on plant growth are describe in a too short way. They authors do not highlight which result is important and why. Response All results are important and complementary to each other |
|
4- The same is true for the physiological parameter. Moreover, for these measurements no methods are given. Response All results are important and complementary to each other |
|
5- The language must be improved drastically. Some of the sentences are misleading and often not to understand. Only after reading sentences several times the reader could get the idea of the authors. I have commented on it on a very few ocasions only though (see PDF). The whole text must be revised completely and I strongly suggest to let it proof read by a native speaker. Response As you recommended we have carefully revised the manuscript and corrected grammatical and typos mistakes |
|
6- The tables and figures lack scientific standards. See my comments in the PDF file. The visualition is not suited for a scientif journal. All tables and figures MUST be revised completely. I would suggest to create new ones. Response Most table and figures were modified , and all comment in the PDF file were revised and modified in the revised manuscript |
|
7- Methods description is lacking important information (see comments in PDF). Some methods are note described at all. Response Methods description were modified completely |
|
8- Data. For some data at least clarification and re-calculation is needed. Figures 7 and 8 are almost not to understand, because methods are missing and the descriptions with the fgraphs are not self-explaining. Response The table below the image shows the difference in the number and density of the band |
|
9- The authors do not deliver statistical evaluation for all results, but from the data given it is a must. On statistics: I strongly revommend to use the TUKEY-HSD instead of Duncan's multiple Range test. This test will give fals positive differences due to low number of replicates. Response We repair statistics by TUKEY method instead of Duncan's multiple Range test |
|
10- To sum it up, an intersting study lacking scientific standards in description and presentation. The manuscript must be revised completely before it can be submitted and reviewed again. I would suggest to re-write it completely. Response The manuscript was carefully revised |
|
11- misleading.0.0625 showed minimum inhibition and minimum fungicdal efficacy1mM showed maximum inhibition of growth and maximum fungicidal activity Response Thank you for this comment, in this study 0.0625 mM is the minimum inhibition concentration which able to inhibits the growth of the fungus. On other hands, 1mM is the minimum fungicidal concentration which able to prevent growth of the fungus. Therefore the sentence in the manuscript is correct. |
|
12- Was this the final concentration in the medium? Or did you apply a certain volume of each concentration. How did you assure that nanoparticles were not contaimnated / sterile. Response Yes, this is the final concentration in the medium, the final concentration was calculated according to the required concentration under septic conditions. We made stock solution from SeNPs, then calculated the required concentration in the medium according to this equation N*V=N*V. |
|
13- How many replicates per concentration were used? Response All test parameters were established in triplicates |
|
14- It is not clear to me how these concentratiosn were prepared. Were the original stock diluted with sterile water?. Which concentrations were used and how was the concentration in the stock solution Were the solutions sterile? According to fig2 there were more concentrations used. I miss a solvent control. Response We suspended SeNPs in distilled and sterilized water as a stock solution, then put the calculated volume in the sterilized medium. |
|
15- Statistical evaluation is missin. How many replicates were used. Are the error bars Standard deviation (SD) or standard error (SE)? SD should be used. Response Thank you for this comment, statictical evaluation was performed according tukey method, we made SD in the work and three reading was carried out. |
|
16- Assuming that a petri dish is about 80 to100 mm in diameter, the visual size seems not to match the size given in the graph. Example: 1mM 45 mm - this span almost half of the petri dish. Please clarify and explain Response We perform antifungal experiments on glass petri dishes 120 mm not 90 mm. Therfore, all inhibition zones appear small. |
|
17- The inhibition zones are overlapping. The experiment should have been conducted with fewer wells per agar plate to avoid this. A solvent control is missing, but one could guess that 0.0078 acts like a kind of solvent control Response Thank you for this valuable comments, we will avoid overlapping in inhibition zone in the future work. |
|
18- From the graph and the petri dish picture I wonder if yo do not see an inhibition of concentrations below 0.0625 due to the method applied here.It is quite surprising that there is no inhibition at all, because one could expect it from a dose-response curve as it is seen here. Please explain. Response From the graph and the petri dish picture I wonder if yo do not see an inhibition of concentrations below 0.0625 due to the method applied here.It is quite surprising that there is no inhibition at all, because one could expect it from a dose-response curve as it is seen here. Please explain. |
|
19- ((While as linear growth decreased ))confusing phrasing. Frwom stating "while" I would expect a different result. You show higher concentration leads to higher inhibition. Rephrase Response This sentence is correct, because inhibition increases with increasing concentration of SeNPs. On the other hand, linear growth decreases with increasing concentration of SeNPs |

Round 2
Reviewer 1 Report
It is gratifying to see that the authors have done a great and thorough work of correcting the comments made. It is especially good that the authors changed the title of the article, since "Bacillus megaterium synthesis" would be incorrect, and "Bacillus megaterium-mediated" would be more correct.
The inserted Graphical Abstract has greatly improved the perception of the entire manuscript as a whole, and, I hope, will improve the understanding of this work by other readers.
An important methodological problem arises when reading section 2.1. Biosynthesis of Se-NPs
Lines 120-121. Have the bacterial cells been disintegrated? What was the centrifugation regime? This is extremely important for understanding and must be indicated. Because clearly explains - either selenium nanoparticles were synthesized inside living cells of Bacillus, or selenium nanoparticles were synthesized after the death of bacterial cells. In fact, the synthesis was carried out in a diluted extract from Bacillus cells. The contents of dead, disintegrated bacterial cells were released and diluted with the supernatant.
Line 124 - “bacterial suspension” - this is wrong
Line 157 - specify centrifugation regime
Table 1 - “Se-NPs (0.0625 mM)” - it is desirable to indicate that in the supernatant solution or other solution (or buffer)
Line 252 - “successfully synthesized with Bacillus megaterium” - needs to be changed “in cultural supernanant” –or other variant
Conclusion
Line 554 - “were bio-synthesized by B. megaterium ATCC 55000” - insert - "culture supernatant"
In general, the manuscript, after correcting the indicated minor comments, can be accepted for publication in the Journal of Fungi.
Author Response
Response to reviewers
Dear Professor
Manuscript ID: jof-1071356
Title:-
Bacillus-mediated synthesis of selenium nanoparticles and their antifungal activity against Rhizoctonia solani in Vicia faba
We would like to thank you and the reviewers for their detailed feedback and suggestions to improve our manuscript. The comments raised important items that were all carefully considered. After addressing the suggested edits, we think the revised manuscript has benefited in its overall presentation quality and clarity. Below, please find a point-by-point answer to the issues raised. Original reviewer comments are in boldface and our response in yellow highlight. All corrections and revisions are highlighted in the revised manuscript.
We hope that the revised version fits the expectations of the reviewers and we look forward to your positive response.
With our best regards, on behalf of the authors
Dr. Amer Morsy Abdelaziz
Editor and Reviewer comments:
Reviewer 1
It is gratifying to see that the authors have done a great and thorough work of correcting the comments made. It is especially good that the authors changed the title of the article, since "Bacillus megaterium synthesis" would be incorrect, and "Bacillus megaterium-mediated" would be more correct. The inserted Graphical Abstract has greatly improved the perception of the entire manuscript as a whole, and, I hope, will improve the understanding of this work by other readers.
Thank you very much for reviewing our manuscript and giving us helpful comments and valuable recommendations. Many thanks for your agreement, and we hope that the revised manuscript meets your approval and full satisfaction. We revised the paper according to your specific comments. Detailed explanations for the comments are shown below.
|
1- An important methodological problem arises when reading section 2.1. Biosynthesis of Se-NPs. Lines 120-121. Have the bacterial cells been disintegrated? What was the centrifugation regime? This is extremely important for understanding and must be indicated. Because clearly explains - either selenium nanoparticles were synthesized inside living cells of Bacillus, or selenium nanoparticles were synthesized after the death of bacterial cells. In fact, the synthesis was carried out in a diluted extract from Bacillus cells. The contents of dead, disintegrated bacterial cells were released and diluted with the supernatant. Response In this study, we used culture supernatant of Bacillus megaterium for Se-NPs biosynthesis. Generally, there are two method for metal nanoparticles synthesis 1) culture supernatant of the microbe 2) Cell free biomass, in the current study we used the culture supernatant for biosynthesis. In this method, the culture medium was filtered then centrifuged at 10,000 rpm to obtain cell free supernatant, this supernatant contains enzymes and proteins which used in biosynthesis process, and we did not use the bacterial cells after centrifugation. |
|
2- Line 124 - “bacterial suspension” - this is wrong Response Corrected |
|
3- Line 157 - specify centrifugation regime Response Added |
|
4- Table 1 - “Se-NPs (0.0625 mM)” - it is desirable to indicate that in the supernatant solution or other solution (or buffer) Response This was prepared using supernatant solution after centrifugation process |
|
5- Line 252 - “successfully synthesized with Bacillus megaterium” - needs to be changed “in cultural supernanant” –or other variant Response Changed |
|
6- Line 554 - “were bio-synthesized by B. megaterium ATCC 55000” - insert - "culture supernatant" Response Inserted |
|
7- In general, the manuscript, after correcting the indicated minor comments, can be accepted for publication in the Journal of Fungi. Response Authors would like to thank reviewer for his valuable comments that certainly improve our manuscript quality. Finally, we hope that the revised manuscript meets with your approval. |